# Socio-Economic Inequalities in Adolescent Summer Holiday Experiences, and Mental Wellbeing on Return to School: Analysis of the School Health Research Network/Health Behaviour in School-Aged Children Survey in Wales

**DOI:** 10.3390/ijerph16071107

**Published:** 2019-03-28

**Authors:** Kelly Morgan, G. J. Melendez-Torres, Amy Bond, Jemma Hawkins, Gillian Hewitt, Simon Murphy, Graham Moore

**Affiliations:** Centre for the Development and Evaluation of Complex Interventions for Public Health Improvement (DECIPHer), School of Social Sciences, Cardiff University, Cardiff CF10 3BD, UK; Melendez-TorresG@cardiff.ac.uk (G.J.M.-T.); BondA3@cardiff.ac.uk (A.B.); HawkinsJ10@cardiff.ac.uk (J.H.); HewittG@cardiff.ac.uk (G.H.); MurphyS7@cardiff.ac.uk (S.M.); MooreG@cardiff.ac.uk (G.M.)

**Keywords:** school holiday, adolescents, family affluence, inequalities, mental wellbeing, mediating effect

## Abstract

The socioeconomic inequalities found in child and adolescent mental wellbeing are increasingly acknowledged. Although interventions increasingly focus on school holidays as a critical period for intervention to reduce inequalities, no studies have modelled the role of summer holiday experiences in explaining socioeconomic inequalities in wellbeing. For this study, we analysed survey data of 103,971 adolescents from 193 secondary schools in Wales, United Kingdom, which included measures of family affluence, experiences during the summer holidays (hunger, loneliness, time with friends and physical activity) and mental wellbeing and internalising symptoms on return to school. Structural equation modelling was used to analyse the data. Although family affluence retained a direct inverse association with student mental wellbeing (*r* = −0.04, *p* < 0.001), 65.2% of its association with mental wellbeing was mediated by the experiences over the summer holidays. FAS score was not directly associated with the student’s self-reports of internalising symptoms (*r* = 0.00, *p* > 0.05). Of all summer holiday experiences, the strongest mediational pathway was observed for reports of loneliness. Although more structural solutions to poverty remain essential, school holiday interventions may have significant potential for reducing socioeconomic inequalities in mental health and wellbeing on young people’s return to school through reducing loneliness, providing nutritious food and opportunities for social interaction.

## 1. Introduction

Childhood and adolescent inequalities have worsened internationally in recent years [1], resulting in adverse psychosocial, physical and educational consequences for young people and families [2]. Socioeconomic inequalities in a range of health risk behaviours as well as mental health and wellbeing emerge and widen throughout childhood and adolescence [3,4,5,6,7]. Following the recent international financial crisis and periods of austerity in many countries, food insecurity [8] (i.e., “insufficient and insecure access to nutritionally adequate food due to resource constraints” (p.523) [9]) is a growing problem. 

Despite having one of the world’s richest economics, the United Kingdom (UK) is among the most unequal countries in the developed world [10]. UK studies have found that half of teachers perceive hunger to be a problem in their school, with most perceiving this problem to have been growing in recent years [11]. The recent estimates indicate that 1 in 5 young people aged under 15 years old live in an environment of moderate to severe food insecurity [12]. The Trussel Trust, which is a network of UK foodbanks, has described recent increases in demand for emergency food for children [13]. Food insecurity has an impact on all members of the family unit. It is linked to depression, poor physical health and reduced school engagement in children and young people [14], while qualitative studies have highlighted that parents report going without food in the school holidays to enable their children to eat, experiencing undernutrition and poor mental wellbeing themselves as a result [15]. 

In many countries, schools take a break of several weeks during the summer months. In the UK for example, most schools break for six weeks. Although this is an opportunity for greater family interaction and positive leisure activities for many, these summer holidays can be stressful periods for low-income families. In many countries, young people from poorer families receive free midday meals during school times and additional meal provision, such as breakfast clubs, have been used in many countries to reduce nutritional inequalities during term time [1]. The absence of these provisions during school holiday periods can add to financial pressures and risk of undernutrition for poorer families [9,12,16,17,18], which have added to difficulties in relation to costs and logistics of childcare [13]. 

A growing body of evidence indicates that young people from poorer backgrounds experience a greater degree of learning loss compared to their peers during the summer break [19]. They take several weeks to catch up with the levels of numeracy and literacy that they achieved prior to the summer holidays while also experiencing periods of isolation, boredom and inactivity [9]. Limited affordable opportunities for school holiday activities and the high cost of childcare can inhibit young people’s opportunities to participate in social activities and interactions [12], which may mean that during the school holidays, young people spend increased amounts of time sedentary and socially isolated [13]. The cost of childcare exceeds many families’ budgets, which often results in working parents leaving paid work or leaving their child without adult supervision during the summer holidays [20], which potentially impacts on their sense of loneliness [20]. Children who are socially disadvantaged have been shown to be at an increased risk of social isolation or loneliness, with limited household and neighbourhood resources making the formation and maintenance of friendships difficult [21,22]. The accumulation of loneliness, hunger and increased sedentary behaviour during summer holiday periods may contribute to poorer health outcomes for children, including mental health and wellbeing [13]. Due to the potential learning loss, these factors may contribute to enduring gaps in educational outcomes between the richest and poorest students [15]. 

The challenges faced by low income families in the holidays have been recognised in the UK and internationally [23,24], with an increase in charitable and government funded school holiday provision, particularly for those families experiencing poverty [2]. Under the Childcare Act 2006, the schools in England and Wales are encouraged to provide affordable and accessible holiday childcare within the existing facilities of schools [25]. An example of such an approach is provided by the ‘Food and Fun’ School Holiday Enrichment Programme (SHEP) in Wales [26]. These initiatives have often been driven primarily from a nutritional perspective, with an explicit aim of tackling holiday hunger. However, they have evolved towards a more holistic focus, including supporting children’s psychosocial wellbeing and opportunities for children to remain physically active and participate in educationally stimulating activities during school holidays. To the author’s knowledge, no study has examined the role of summer holiday experiences, such as hunger, loneliness, time spent with friends and physical (in)activity, in explaining socioeconomic differences in mental health and wellbeing on return to school after the summer break. 

This study draws upon data from a large national survey in Wales to examine:The association of socioeconomic status with mental wellbeing and internalising symptoms in adolescents on return to school after the summer holidays;Associations of socioeconomic status with summer holiday experiences of hunger, loneliness, time spent with friends and physical activity;Associations of summer holiday experiences of hunger, loneliness, time with friends and physical activity with mental wellbeing and internalising symptoms on return to school;The relative contributions of summer holiday experiences of hunger, loneliness, time with friends and physical activity in mediating the association between socioeconomic status and mental wellbeing.

## 2. Materials and Methods 

### 2.1. Participants, Sampling and Data Collection

The data used in this study were collected from September–December 2017. The School Health Research Network (SHRN) Student Health and Wellbeing Survey was completed by 103,971 secondary school students aged 11–16 years old who were recruited from 193 secondary schools (including four private schools) across Wales. Schools were recruited to SHRN either via their participation in the Welsh Health Behaviour in School-Children (HBSC) survey in 2013 or via two subsequent rounds of open recruitment in 2015 and 2017, including all maintained secondary schools in Wales as of 2017. For the first time in 2017, he HBSC survey was nested within the larger SHRN survey and is part of a wider World Health Organization (WHO) Cross-National survey that involves 49 countries. Schools were asked to survey all pupils where possible and to survey mixed ability classes so as to minimise biases where not all classes were included. Participants were assured of anonymity and confidentiality and data collection took place within the classroom environment. The survey was voluntary and completed anonymously by the majority of students. In 39 schools, the students were asked to provide some identifiable information at the end of the survey as part of a data linkage pilot project. The students were informed that their identifying information would be stored separately to their survey responses, used for research only and never used to identify their responses and were not excluded from the survey if they chose not to provide this. Only the fully anonymised data are used for the analyses in this paper. The total sample analysed in this paper consisted of the 101,910 students across 193 schools after the exclusion of pupils who did not answer the question on their sex.

### 2.2. Measures 

#### 2.2.1. Socioeconomic Status

The Family Affluence Scale (FAS [27,28]) was used as an indicator of child material affluence. A sum score is derived from 6 survey items, which asked; (1) “Do you have your own bedroom?” (2) “How many computers does your family own?” (3) “Does your family own a car, van or truck?” (4) “Does your family have a dishwasher at home?” (5) “How many bathrooms are in your home?” and (6) “How many times did you and your family travel out of Wales on holiday/vacation last year?” The scores for each item were summed to give a total affluence score, which was dichotomised via a median split into high and low affluence.

#### 2.2.2. Summer Holiday Experiences

The participant experiences over the most recent summer holidays were derived from four survey items, which asked about the frequency of the following experiences: (a) Spend time with friends, (b) Exercise in your free time so much that you got out of breath or sweated, (c) Go to bed hungry because there wasn’t enough food in the house and (d) Feel lonely (responses: (1) “None of the time”, (2) “Rarely”, (3) “Some of the time”, (4) “Often”, (5) “All of the time” and (6) “I do not want to answer”). Each survey item was used as a mediator in the analysis.

#### 2.2.3. Sociodemographic Variables 

Participants reported their age (year and month of birth), gender (boy/girl) and ethnicity (response options: “White British”; “White Irish, White Gypsy/Traveller”; “White Other”; “Mixed or Multiple Ethnic Group”; “Pakistani”; “Indian”; “Bangladeshi”; “Chinese”; “African”; “Caribbean or Black”; “Arab”; “Other”. Those reporting White were categorised as 0 and all other responses coded as 1 (referred to as Black Minority Ethnic (BME)). 

#### 2.2.4. Student Mental Well Being 

(a) Short Warwick-Edinburgh Mental Wellbeing Scale (SWEMWBS)

SWEMWBS [29], which is an indicator of child mental wellbeing, was derived from seven survey items, which asked about the following experiences over the last two weeks: a) I’ve been feeling optimistic about the future, b) I’ve been feeling useful, c) I’ve been feeling relaxed, d) I’ve been dealing with problems well, e) I’ve been thinking clearly, f) I’ve been feeling close to other people, g) I’ve been able to make up my own mind about things (responses: (1) “None of the time”, (2) “Rarely”, (3) “Some of the time”, (4) “Often”, (5) “All of the time” and (6) “I do not want to answer”). The responses were summed to form a scale score [30].

(b) Internalising Symptoms 

As an indicator of depression and anxiety symptomatology [31], the participants were asked about the frequency of which they had felt each of the following over the past six months: (a) Feeling low, (b) Irritability or bad temper, (c) Feeling nervous and (d) Difficulties in getting to sleep (responses: (1) “About every day”, (2) “More than once a week”, (3) “About every week”, (4) “About every month”, (5) “Rarely or never” and (6) “I do not want to answer”). The responses were summed to form a scale score.

### 2.3. Data Analyses

We first undertook descriptive analyses on included variables. After this, to account for the missing data, we used multiple imputation with chained equations. The imputations used a logit link for categorical variables and predictive mean matching for scale scores, which included grade and sex as auxiliary variables. The imputations were carried out in Stata v 14 (Statacorp, College Station, TX, USA). Complete-case analyses were conducted as a sensitivity analysis and as there was minimal difference between complete case analyses and those using multiple imputation, only the latter are reported. We specified and estimated two path analyses: one estimates the mediating effects of summer holiday experiences on the relationship between family affluence and mental wellbeing while the other estimates the mediating effects of summer holiday experiences on the relationship between family affluence and internalising symptoms. Both path analyses included the direct effect of family affluence on outcomes as well as theoretically informed covariances between the disturbances of the mediators. All path analyses were estimated using weighted least squares to account for the use of ordinal variables in the analysis. Path analyses were estimated on each imputed dataset before the parameter estimates were combined using Rubin’s rules. The indirect effect was also expressed as the percentage of the total direct effect captured by the mediators. Analyses were carried in R using -lavaan- (Rosseel, 2014, Auckland, New Zealand).

### 2.4. Ethics

Schools registered to participate in the study before the parents in each school were sent a letter informing them about the survey and they had the option of withdrawing their child from data collection. Additionally, at the start of each data collection session, the participants were asked to provide consent after having read through the screens of information at the start of the online survey and having had the study explained by their teachers. All young people had the opportunity to withdraw from the data collection session at any time. The study was approved by Cardiff University School of Social Sciences Research Ethics Committee (SREC/2190).

## 3. Results

### 3.1. Descriptive Statistics

A total of 101,910 young people were included in the main analysis (Table 1). Overall, 49.5% were males and the mean (± SD) age of the respondents was 13.2 years (± 1.5). The mean (± SD) score for SWEMWBS and internalising symptoms of respondents was 22.0 (± 4.5) (range 7–35) and 5.7 (± 4.4) (range 0–16) respectively. The prevalence of young people experiencing hunger at bedtime or feelings of loneliness at least some of the time was 12.9% and 30.4%, respectively. Overall, 26.0% and 16.1% of young people reported that they rarely or never spent time exercising or with friends.

Table 2 provides a breakdown of responses according to FAS. As shown, there was a strong trend across each summer holiday experience, with those from a low FAS reporting more frequent adverse experiences.

### 3.2. Associations between Summer Holiday Experiences and Socioeconomic Status

As shown in Figure 1 and Figure 2, a low FAS was associated with all reported summer holiday experiences. There was a positive association of a low FAS with the frequency of reported hunger at bedtime (r = 0.07, *p* < 0.001) and feeling lonely (r = 0.10, *p* < 0.001). A low FAS had a direct negative association with spending time with friends (r = −0.09, *p* < 0.001) and time spent exercising (r = −0.14, *p* < 0.001).

The covariances estimated between the reports of loneliness and bedtime hunger (r = 0.39, *p* < 0.001) and between loneliness and spending time with friends (r = 0.25, *p* < 0.001) were positive and statistically significant. Conversely, the covariance estimated between reports of loneliness and exercise was negative and significant (r = −0.15, *p* < 0.001). The covariance estimated between time spent exercising and time spent with friends was positive and significant (r = 0.29, *p* < 0.001).

### 3.3. Student’s Mental Wellbeing 

The overall fit of the structural model predicting students’ mental wellbeing was good based on the standard fit criteria (SWEMWBS: over 10 imputations, χ2 ps > 0.34; RMSEA = 0.000–0.001). The results of the SEM are depicted in Table 3. 

The indirect effects of FAS on student mental wellbeing are realised through each of the summer holiday experience variables. The path estimates between the frequency of feeling lonely and mental wellbeing (r = −0.38, *p* < 0.001) and between hunger and mental wellbeing (r = −0.11, *p* < 0.001) were both statistically significant, with more frequent experiences of both associated with poorer mental wellbeing. Similarly, the pathways between the frequency of exercise and mental wellbeing (r = 0.18, *p* < 0.001) and frequency of spending time with friends and mental wellbeing (r = 0.05, *p* < 0.001) were significant. The students reporting more frequent exercise or time with friends over the summer holidays were more likely to report positive mental wellbeing. 

Finally, a low FAS had a direct effect on student mental wellbeing (r = −0.04, *p* < 0.001). Collectively, 65.2% of the effect of FAS on mental wellbeing was mediated by the experiences over the summer holidays.

### 3.4. Internalising Symptoms 

The overall fit of the structural model in predicting the students’ internalising symptoms score was good based on the standard fit criteria (over 10 imputations, χ2 ps > 0.36; RMSEA = 0.000–0.001). Table 4 displays all standardised path coefficient estimates and Figure 2 displays the full SEM.

The pathways between the frequency of feeling lonely and reports of internalising symptoms (r = 0.55, *p* < 0.001) and between going to bed hungry due to a lack of food and internalising symptoms (r = 0.05, *p* < 0.001) were both statistically significant. The pathway between the frequency of spending time with friends and internalising symptoms (r = 0.01, *p* < 0.001) was small yet significant but not in the expected direction. The pathway between the frequency of exercise and internalising symptoms (r = −0.09, *p* < 0.001) was also significant, which indicated that an increased frequency of exercise was associated with decreased internalising symptoms. 

As shown in Figure 2, FAS was not directly associated with the students’ reports of internalising symptoms (r = 0.00, *p* > 0.05). That is to say, the data are consistent with a hypothesis that the association of FAS with internalising symptoms was completely mediated by experiences over the summer holidays. Hence, while the relationship between FAS and each of the summer holiday variables is indicative of better experiences for more affluent families, the direct effect of FAS on internalising symptoms indicates that FAS is not associated with internalising symptoms beyond the effect through summer holiday experiences.

Among both structural models, the strongest mediational pathway was observed for reports of loneliness out of all summer holiday experiences.

## 4. Discussion

Consistent with past research [3,4,7,32,33], we observed that a low socioeconomic status was associated with poorer wellbeing among adolescents in Wales. However, while previous studies have shown that socioeconomic status is associated with adolescent mental health and wellbeing, ours is, to our knowledge, the first study to examine the mediating role of experiences during the summer holiday period (such as hunger, loneliness, time with friends and participation in physical activity) in explaining socioeconomic inequalities in wellbeing on return to school. This is despite a growing emergence of interventions, which aim to reduce socioeconomic inequality through targeting these mechanisms [18]. 

Nearly one in six young people reported frequent experiences of loneliness during the summer holidays. We observed a significantly higher risk of loneliness among young people from poorer families, with over a fifth experiencing loneliness at least “often” or “all of the time” (compared to one in seven and one in nine peers reporting a medium and high FAS, respectively). This finding is consistent with research evidence that many disadvantaged children from working families are often left for long periods of time throughout the holidays due to the challenges in meeting costs of childcare while limited affordable activities for young people to engage in may limit opportunities for social interaction [20]. 

Further, 1 in 6 young people reported that they “never” or “rarely” spend time with friends throughout the summer months, which was again significantly higher among young people from poorer families (i.e., 21% compared to 15% and 12% of peers reporting a medium and high FAS correspondingly). Although a number of studies have demonstrated the protective role of social relationships and support for wellbeing and mental health throughout the human lifespan [34,35,36], several studies have also demonstrated how loneliness during childhood can precede depression in adolescence in addition to increased healthcare usage and associated concerns about health [37,38]. Hence, summer holidays represent a potentially critical period during which loneliness may adversely impact young people’s mental wellbeing.

Approximately 1 in 16 young people reported that they “often” or “always” went to bed hungry during the summer holiday period. Other international studies have reported relatively higher estimates of food insecurity at around 11% [39,40] although the methods for assessing food insecurity vary widely. In the present study, there was a trend for higher reports among children from poorer families (e.g., 7.2%, 5.8% and 5.1% for those reporting a low, medium and high FAS, respectively). Previous studies have shown that the children who report going to bed hungry are also more likely to report lower consumption of fruits and vegetables, higher intakes of convenience and fried foods and more frequent reports of emotional and physical symptoms [41]. Compounding the adverse effects of loneliness already described, the literature suggests that children from food insecure families are also less likely to socialise outside of school and more likely to fall behind academically [42,43], which is consistent with the covariances between hunger and loneliness observed in this study. 

The declines in physical activity have previously been observed during the summer holiday period [44]. Over a quarter of young people in this study reported never or rarely engaging in exercise during the summer holiday period. Of these, a greater proportion of young people were from poorer families (e.g., a third of those in the poorest tertiles vs just under one in five in the most affluent). Our data also indicated the covariance of loneliness with a lower level of physical activity as participation in exercise during the summer holidays was also significantly lower among children from poorer families. The time spent in exercise was also inversely associated with socioeconomic status and independently associated with wellbeing on return to school.

Our study provides evidence that all of the measured summer holiday experiences were significantly associated with young people’s wellbeing and mental health outcomes. The one notable exception was a positive association between time with friends and internalising symptoms as this suggests that such symptoms increase when the time with friends was higher. Our previous research has shown that time with friends can positively or negatively predict mental health and wellbeing, which partially depends on whether it occurs within the context of supportive family relationships [31]. However, this coefficient was small (0.01) and most likely a consequence of the very large analysis sample. Of all summer holiday experiences, the reports of loneliness had the greatest association with the reports of poorer wellbeing and mental health. Hence, one of the most important mechanisms of school holiday intervention in terms of reducing inequalities is likely to be reducing experiences of loneliness. 

### Limitations and Implications for Future Research

This study benefitted from a large nationally representative sample and use of robust statistical methods. Nevertheless, it suffers a number of important limitations. It relies exclusively on self-reported data and may be subject to recall biases and social desirability. Although our recent analysis indicates that SWEMWBS has acceptable measurement invariance properties and validity in this age group [30], our measure of internalising symptoms is as yet not validated. Our measure of socioeconomic status that was related to indirect markers of family affluence and alternative markers of SES, such as household income or parental education, may have revealed differing associations. That said, compared to other family affluence measures relying on parental occupation, education and/or income, previous research indicates that the FAS has superior criterion validity and is less affected by nonresponse bias [27]. Nevertheless, it is an imperfect measure of socioeconomic status and triangulation with other measures if they were available would have helped us test the validity of our findings. Our indicator of food insecurity was related to ‘hunger’ as opposed to the quality of foods consumed. It is likely that many poorer families provide their children with sufficient calories during the summer holidays via cheaper foods of lower nutritional value. Study data were collected from pupils and not from whole family units. Although we observed a relatively weak association between socioeconomic status and child holiday hunger, we cannot rule out the possibility that parents are skipping meals to provide for their children as shown in earlier research [8]. Consequently, an indicator of hunger at the child level likely underestimates household food insecurity. The cross-sectional nature of this study means that the directionality of relationships cannot be firmly established. Indeed, potential reverse causality could account for some of the findings in the present study. For instance, poor mental wellbeing may promote more loneliness, less exercise and lack of appetite. That said, there is a degree of implicit temporality among our data, with socioeconomic status being relatively stable over time, each mediator relating to the summer holiday period (months of July and August) and our mental wellbeing outcome (SWMWEBS) relating to the two weeks immediately preceding reporting and our measure of internalising symptoms to the past 4 weeks. The priorities for future research include prospective re-examination of trends identified here using longitudinal data and the development and robust evaluation of summer holiday interventions to improve pupil mental health and wellbeing. Planned future data linkage will also enable us to explore impacts on educational attainment, which were beyond the remit of this paper.

## 5. Conclusions

Our analyses are the first to examine the importance of summer holiday experiences, such as hunger, loneliness, participation in exercise and time with friends as mediators of young people’s mental wellbeing on return to school. Summer holiday experiences appear to explain a large proportion of the relationship between socioeconomic status and mental health and wellbeing. Taken together, the present findings suggest that school holiday interventions through reducing loneliness, providing nutritious food and opportunities for social interaction may offer significant potential for reducing socioeconomic inequalities in mental health and wellbeing on young people’s return to school. It is important to recognise that school holiday interventions provide a short term fix. The socioeconomic inequalities in health are a consequence of structural inequalities within society and hence structural solutions, albeit likely to require considerable investment [45], are vital in preventive efforts going forward.

## Figures and Tables

**Figure 1 ijerph-16-01107-f001:**
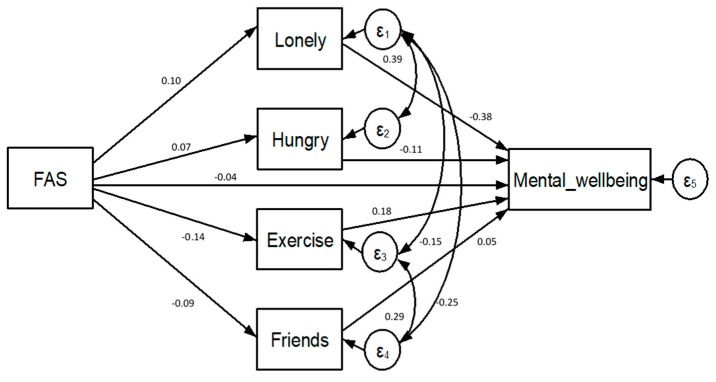
Model 1, which was the Short Warwick-Edinburgh Mental Wellbeing Scale (SWEMWBS).

**Figure 2 ijerph-16-01107-f002:**
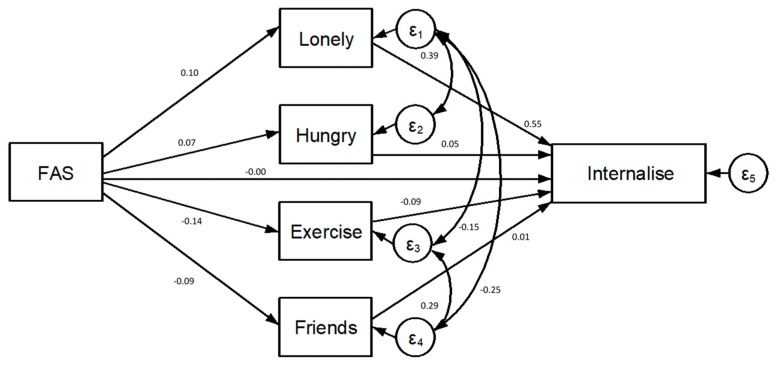
Model 2, which focused on internalising symptoms.

**Table 1 ijerph-16-01107-t001:** Demographic and socioeconomic characteristics of the study participants.

Personal Characteristic (N)	Variable	Percent (%)
Gender (101,901)	Male	49.51
	Female	50.49
School year (101,910)	7	21.90
	8	21.63
	9	21.27
	10	18.83
	11	16.37
FAS (101,910)	Low	34.91
	Medium	31.42
	High	33.67
Summer holiday experiences		
Going to bed hungry (100,026)	None of the time	67.65
	Rarely	19.48
	Some of the time	6.81
	Often	3.62
	All of the time	2.44
Loneliness (99,650)	None of the time	42.57
	Rarely	27.02
	Some of the time	14.92
	Often	9.24
	All of the time	6.24
Spend time with friends (101,908)	None of the time	4.80
	Rarely	11.33
	Some of the time	20.08
	Often	35.80
	All of the time	28.00
Exercise in free time (99,804)	None of the time	7.89
	Rarely	18.10
	Some of the time	29.12
	Often	30.87
	All of the time	14.01
Mental health indicators		
SWEMWBS (101,910) ^		22.0(4.45) (7–35)
Internalising symptoms (101,910) ^		5.75(4.38) (0–16)

^ Mean (SD) (Range).

**Table 2 ijerph-16-01107-t002:** Summer holiday experiences according to Family Affluence Scale (FAS) score.

Variable	FAS	Responses				
**Going to Bed Hungry**		**None of the Time**	**Rarely**	**Some of the Time**	**Often**	**All of the Time**
	Low	22,232 (64.03)	7196 (20.73)	2790 (8.55)	1511 (4.35)	990 (2.85)
	Medium	21,301 (67.69)	6239 (1.98)	2106 (6.69)	1082 (3.44)	739 (2.35)
	High	24,136 (71.32)	6055 (17.89)	1913 (5.65)	1023 (3.02)	713 (2.11)
Loneliness						
	Low	13,428 (38.82)	8771 (25.36)	5466 (15.80)	3976 (11.50)	2946 (8.52)
	Medium	13,418 (42.83)	8519 (27.19)	4759 (15.19)	2826 (9.02)	1805 (5.76)
	High	15,574 (46.16)	9640 (28.57)	4645 (13.77)	2405 (7.13)	1472 (4.36)
Spend time with friends						
	Low	2478 (7.05)	4828 (13.74)	7336 (20.88)	11381 (32.39)	9111 (25.93)
	Medium	1318 (4.15)	3557 (11.21)	6366 (20.07)	11569 (36.47)	8912 (28.09)
	High	1048 (3.08)	3043 (8.94)	6560 (19.26)	13173 (38.68)	10228 (30.04)
Exercise in free time						
	Low	3966 (11.45)	7530 (21.74)	10,223 (29.52)	9120 (26.33)	3794 (10.95)
	Medium	2270 (7.23)	5605 (17.86)	9530 (30.37)	9705 (31.07)	4272 (13.61)
	High	1638 (4.85)	4929 (14.59)	9313 (27.56)	11989 (35.48)	5920 (17.52)

**Table 3 ijerph-16-01107-t003:** Standardised path coefficient estimates of Family Affluence Scale (FAS) score and summer holiday experiences on student mental wellbeing (SWEMWBS score).

Parameter	Standard Estimate	z−Value
FAS		
Loneliness	0.10	28.80
Hunger	0.07	19.04
Exercise	−0.14	−43.34
Time with Friends	−0.09	−28.23
SWEMWBS on		
FAS	−0.04	−14.11
Loneliness	−0.38	−122.78
Hunger	−0.11	−28.39
Exercise	0.18	61.30
Time with Friends	0.05	16.82
Covariances		
Lonely with		
Friends	−0.25	−76.70
Hungry	0.39	109.52
Exercise	−0.15	−44.91
Exercise with friends	0.29	95.41

**Table 4 ijerph-16-01107-t004:** Standardised path coefficient estimates of FAS and summer holiday experiences on internalising symptoms.

Parameter	Standard Estimate	z−Value
FAS		
Lonely	0.10	28.83
Hungry	0.07	19.00
Exercise	−0.14	−43.51
Friends	−0.09	−28.31
Internalising symptoms on		
FAS	−0.00	−0.65
Lonely	0.55	136.54
Hungry	0.05	13.84
Exercise	−0.09	−30.54
Friends	0.01	3.21
Covariances		
Lonely with		
Friends	−0.25	−76.69
Hungry	0.39	109.54
Exercise	−0.15	−45.46
Exercise with friends	0.29	95.77

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
