# Peer review of "Socio-Economic Inequalities in Adolescent Summer Holiday Experiences, and Mental Wellbeing on Return to School: Analysis of the School Health Research Network/Health Behaviour in School-Aged Children Survey in Wales"

_ijerph, 2019, doi:10.3390/ijerph16071107_

Round 1

Reviewer 1 Report

This is a well written manuscript addressing an important question regarding activity, economic status and children's mental health and performance in school.  However, there are some questions regarding one of the tests used. In addition, there are some other variables that the authors may be able to include in their analyses that would provide additional information for determine what are the most important factors influence a child's health and performance in school.

1)  There are some questions regarding the validity and reliability of the FAS scale.  Can this be addressed in the discussion.

2) Are there data on where these students live.  For example, are they located close to school and friends?  Is there an association between geographical location and any of the reported measures?

3) The authors state that one of the reasons for performing this study was to look at how summer activity and food availability affected back to school performance.  Are there data that correlate any of the variable examined with measures of academic performance after returning to school?

Author Response

Response to Reviewer 1 Comments

Point 1: There are some questions regarding the validity and reliability of the FAS scale.  Can this be addressed in the discussion?

Response 1: We have inserted the following text: “That said, compared to other family affluence measures relying on parental occupation, education and/or income, previous research indicates that the FAS has superior criterion validity and is less affected by nonresponse bias [27]. Nevertheless, it is an imperfect measure of socioeconomic status, and triangulation with other measures were they available would have helped us test the validity of our findings.” (Lines 348-353)

Point 2: Are there data on where these students live.  For example, are they located close to school and friends?  Is there an association between geographical location and any of the reported measures?

Response 2: While the reviewer raises some interesting questions for future exploration, these were not questions we sought to address with our analysis.

Point 3: The authors state that one of the reasons for performing this study was to look at how summer activity and food availability affected back to school performance. Are there data that correlate any of the variable examined with measures of academic performance after returning to school?

Response 3: We do not have these measures available to us and this was not a question that we set out to address. However, to help address this point, we have inserted the following sentence within our paper, “Planned future data linkage will also enable us to explore impacts on educational attainment which were beyond the remit of this paper.” (Line 369)

Reviewer 2 Report

The matter of the manuscript is really interesting and have a direct impact in terms of health and social policies. It is a great research paper with clear results and methodology. Congratulations! However, could the authors have a look to these three little questions?

1.- Could you add a link to the reference 13?

2)  p.87 Which studies? Reference them.  “To date, few studies have examined the role of summer holiday experiences such as hunger, loneliness, time spent with friends and physical (in)activity, in explaining socioeconomic differences in mental health and wellbeing on return to  school after the summer break.

3) Why did you select the SWEMWBS? You have also the international “Strengths and difficulties questionnaire”(SDQ), created in London-UK (Goodman, 1997)*. It measures at the same time internalising symptoms and global mental health. It has demonstrated an international used and validation. Why did you discard this instrument? What do you think about this questionnaire? (see  www.sdqinfo.com)

 *Goodman, R. (1997). The strengths and difficulties questionnaire: A research note. Journal of Child Psychology and Psychiatry and Allied Disciplines, 38(5), 581–586. https://doi.org/10.1111/j.1469-7610.1997.tb01545.x

Author Response

Response to Reviewer 2 Comments

Point 1: Could you add a link to the reference 13?

Response 1: Thank you, a link has now been inserted.

Point 2: p.87 Which studies? Reference them.  “To date, few studies have examined the role of summer holiday experiences such as hunger, loneliness, time spent with friends and physical (in)activity, in explaining socioeconomic differences in mental health and wellbeing on return to  school after the summer break. “

Response: Thank you for identifying this sentence. For accuracy, this sentence has been amended to read “To the author’s knowledge, no study has examined …” (Line 87)

3) Why did you select the SWEMWBS? You have also the international “Strengths and difficulties questionnaire”(SDQ), created in London-UK (Goodman, 1997)*. It measures at the same time internalising symptoms and global mental health. It has demonstrated an international used and validation. Why did you discard this instrument? What do you think about this questionnaire? (see  www.sdqinfo.com)

Response: Thank you for your comments. The SDQ, while widely used to measure adolescent mental health, was not included within the survey data which formed the basis of our analysis. We acknowledge that our measure of internalising symptoms is as yet unvalidated. This is currently reflected in the following text: “While our recent analysis indicates that SWEMWBS has acceptable measurement invariance properties and validity in this age group [30], our measure of internalising symptoms is as yet not validated.” (Line 344)

Reviewer 3 Report

 Thank you for opportunity for reviewing this paper “Socio-economic inequalities in adolescent summer 2 holiday experiences, and mental wellbeing on return 3 to school: analysis of the School Health Research 4 Network / Health Behaviour in School-aged Children 5 survey in Wales”. The contents of the paper are very important and interesting for both teachers and health care providers to consider when they want to design summer holiday interventions for school-aged children. My small comment is that based on your literature reviewing, you may add some hypotheses for the study.

Author Response

Response to Reviewer 3 Comments

Point 1: My small comment is that based on your literature reviewing, you may add some hypotheses for the study.

Response 1: Thank you. Your comment indicates a preference for explicit hypotheses, whereas we have framed these within the text as research questions. Given that this is largely a matter of stylistic preference, and this has been offered as a suggestion rather than a necessary improvement, we have chosen not to alter the manuscript.